# A Robust Thermal Infrared Vehicle and Pedestrian Detection Method in Complex Scenes

**DOI:** 10.3390/s21041240

**Published:** 2021-02-10

**Authors:** Yang Liu, Hailong Su, Cao Zeng, Xiaoli Li

**Affiliations:** 1Key Laboratory of High-Speed Circuit Design and EMC Ministry of Education, Xidian University, Xi’an 710071, China; liuyang@mail.xidian.edu.cn (Y.L.); hailongsu@stu.xidian.edu.cn (H.S.); 2National Laboratory of Radar Signal Processing, Xidian University, Xi’an 710071, China; 3Key Laboratory of Intelligent Perception and Image Understanding of Ministry of Education, Xidian University, Xi’an 710071, China; LiXiaoli@stu.xidian.edu.cn

**Keywords:** vehicle and pedestrian detection, thermal infrared images, the optimized FSAF module, motion blur

## Abstract

In complex scenes, it is a huge challenge to accurately detect motion-blurred, tiny, and dense objects in the thermal infrared images. To solve this problem, robust thermal infrared vehicle and pedestrian detection method is proposed in this paper. An important weight parameter *β* is first proposed to reconstruct the loss function of the feature selective anchor-free (FSAF) module in its online feature selection process, and the FSAF module is optimized to enhance the detection performance of motion-blurred objects. The proposal of parameter *β* provides an effective solution to the challenge of motion-blurred object detection. Then, the optimized anchor-free branches of the FSAF module are plugged into the YOLOv3 single-shot detector and work jointly with the anchor-based branches of the YOLOv3 detector in both training and inference, which efficiently improves the detection precision of the detector for tiny and dense objects. Experimental results show that the method proposed is superior to other typical thermal infrared vehicle and pedestrian detection algorithms due to 72.2% mean average precision (mAP).

## 1. Introduction

Thermal infrared object detection technology, due to its noncontact and passive detection characteristics, overcomes the shortcomings of radar object detection technology’s poor portability and the difficulty in detecting the objects in radar blind areas. It has broken through the limitations of optical object detection technology, which has a poor capability to penetrate smoke and dense fog and cannot work all day, and has become a research hotspot in the fields of military reconnaissance, traffic management, and autonomous driving.

With the improvement of the level of thermal infrared detectors and the computing power of graphics processing unit (GPU), the algorithms for thermal infrared object detection and recognition continue to emerge. The previous traditional infrared object detection algorithms generally used the strategy of sliding windows to select the region proposal [1,2,3,4,5], and then used the algorithm based on handcrafted features to extract the shape or texture features of the objects, such as scale-invariant feature transform (SIFT) [6] and histogram of oriented gradient (HOG) [7]. Finally, the support vector machine (SVM) [8] classification algorithm was employed to complete the object category recognition. However, the strategy of sliding windows will generate a large number of redundant boxes, and the computational complexity of feature extraction is high. At the same time, the extraction algorithms based on handcrafted features are difficult to capture the higher-level semantic information of the objects, resulting in a low detection precision.

As the performance of handcrafted features reaches the limit, the development of object detection technology reached a period of stagnation after 2010. Until 2012, the success of AlexNet [9] in the ImageNet Large Scale Visual Recognition Competition [10] proved that the features extracted by deep convolutional neural networks are more powerful than handcrafted features. This is because a deep convolutional neural network can learn more advanced and complex feature representation of an image; then, a natural question is whether we can bring it to object detection. R. Girshick et al. took the lead to break the deadlocks in 2014 by proposing the regions with CNN features (RCNN) [11] for object detection. It starts with the extraction of a set of object proposals by selective search [12]. Then, each proposal is rescaled to a fixed size image and fed into a CNN model trained on ImageNet to extract features. Finally, linear SVM classifiers are used to predict the presence of an object within each region and to recognize object categories. RCNN successfully solved the problems of the high computational complexity of feature extraction and the low detection precision of traditional object detection methods, and proved that the object detection algorithms based on deep convolutional neural networks have excellent robustness to the variation of object diversity. Therefore, almost all subsequent object detection methods with excellent performance are based on deep convolutional neural networks. These methods can be divided into two categories; the first uses the one-stage object detection method for end-to-end network training and prediction. For instance, you only look once (YOLO) [13] calculates the global feature map for location and category prediction at the fully connected layer. The single shot multibox detector (SSD) [14] uses the multi-scale feature map generated by the deep network layered down sampling structure for prediction. The second uses the two-stage method to detect objects, such as Faster R-CNN [15] algorithm, which generates the region of interest from the feature map extracted from the preprocessed network in its first stage, then selects the features of the region of interest from the shared feature map to predict more accurate classification and location in its second stage.

The above two categories of algorithms are mainly used in the detection of optical objects. However, infrared images have a lower resolution than optical images and are susceptible to ambient temperature, especially for some tiny objects that only occupy a few pixels in the infrared images, which brings a huge challenge to infrared object detection. Moreover, it becomes quite difficult to accurately locate the infrared objects in the situation where the objects are dense and mutually occluded. Aiming at the above problems, Yi et al. [16] combined deep convolutional neural networks and image preprocessing methods to enhance contrast and details, and proposed a night object recognition method based on infrared thermal imaging and YOLOv3 [17]. However, this method has poor detection performance for tiny objects that are difficult to model based on anchor boxes, and the object recall is low. Zhang et al. [18] proposed vehicle detection in the aerial infrared images via an improved YOLOv3 network. The algorithm expands the anchor boxes to four scales and constructs a new structure of the improved YOLOv3 network with only 16 layers, which improves the detection precision of the small vehicles. Nevertheless, this method still cannot accurately locate the dense vehicles.

Besides, motion blur of the object is another huge challenge to infrared object detection. Motion blur is mainly caused by the relative movement between the camera, the object, and the background, which will lead to strong noise and background clutter interference in the infrared imaging system, making the object signal relatively weak. More specifically, the problem of motion blur will make thermal infrared objects lack effective shape and texture features, resulting in a sharp decrease in the robustness of the detector.

To solve the above problems, this paper proposes a robust thermal infrared vehicle and pedestrian detection method. Specifically speaking, an important weight parameter β is first proposed to reconstruct the loss function of the feature selective anchor-free (FSAF) [19] module in its online feature selection process; the FSAF module is optimized to enhance the detection performance of the detector for motion-blurred objects. The proposal of parameter provides an effective solution to the challenge of motion-blurred object detection. Then, the optimized anchor-free branches of the FSAF module are plugged into the YOLOv3 single-shot detector with a similar feature pyramid network (FPN) [20] structure, and work jointly with the anchor-based branches of the YOLOv3 detector in both training and inference, which efficiently enhances the detection performance of the detector for tiny and dense objects. In the end, two effective strategies, Complete Intersection over Union (CIoU) [21] loss and the Soft-NMS [22] algorithm are adopted to further increase the detection precision of multi-category objects. Compared with some typical infrared object detection methods such as SSD, Faster R-CNN, and YOLOv3, the method proposed increases the mean average precision (mAP) scores by 12.6%, 8.1%, and 6.4%, respectively.

The rest of this paper is organized as follows. Section 2 details the proposed method. Section 3 conducts the experiments based on the infrared image dataset to verify the effectiveness of the proposed method. Section 4 summarizes this paper.

## 2. Algorithm Process

The algorithm’s general structure is the optimized FSAF module that is plugged into the YOLOv3 network architecture, as shown in Figure 1. The input data of the detector is the original infrared image, and the backbone network Darknet-53 [17] of the YOLOv3 detector is used to complete the feature extraction of each instance information in the original image. Then, the anchor-free branches of the optimized FSAF module are added to the 79th, 91st, and 103rd layers of the YOLOv3 feature pyramid layer, respectively. Moreover, the optimized FSAF module works jointly with the anchor-based branches of the YOLOv3 detector. Finally, the Soft-NMS algorithm is adopted to select the final object detection boxes and obtain the infrared vehicle and pedestrian detection results, which are the output data. The basic concept of the Soft-NMS algorithm is introduced in Section 2.4.

### 2.1. The FSAF Module

The general concept of the FSAF module is online feature selection applied to the training of multi-level anchor-free branches, and it addresses two limitations brought up by the conventional anchor-based detection: (1) heuristic-guided feature selection; (2) overlap-based anchor sampling. The basic concept of overlap-based anchor sampling is that instance boxes are matched to anchor boxes based on the Intersection over Union (IoU) overlap, and each instance is always matched to the closest anchor box(es). The general concept of each instance refers to each target in the infrared image, for example, different pedestrians represent different instances in the image. The general concept of heuristic-guided feature selection is that anchor boxes are associated with a certain level of feature map by human-defined rules, such as box size. Therefore, the selected feature level for each instance is purely based on ad hoc heuristics. The motivation of the FSAF module is to let each instance select the best level of feature freely to optimize the network, so there should be no anchor boxes to constrain the feature selection in the module. Instead, the FSAF module encodes the instances in an anchor-free manner to learn the parameters for classification and regression. Besides, the FSAF module calculates the sum of classification loss and regression loss for each feature level of all anchor-free branches in the online feature selection process. The classification loss is focal loss [23], the regression loss is Intersection over Union (IoU) loss [24]. Then, the best pyramid level $l^{*}$ yielding the minimal summation of losses is selected to learn the instance, i.e.,
(1)$$l^{*}=\arg\underset{l}{\min}L_{FL}^{I}(l)+L_{IoU}^{I}(l),\;l=1,2,3$$
(2)$$L=\underset{}{\min}\{L_{FL}^{I}(l)+L_{IoU}^{I}(l)\},\;l=1,2,3$$
where $L_{FL}^{I}(l)$ and $L_{IoU}^{I}(l)$ are classification loss and box regression loss on the pyramid level, respectively. I is the instance, l is the pyramid level, and its value can be 1, 2, or 3, and L is the minimal summation of $L_{FL}^{I}(l)$ and $L_{IoU}^{I}(l)$.

The three pyramid levels refer to the three different scale feature maps of the YOLOv3 network, which correspond to the 79th, 91st, and 103rd layers of the YOLOv3 feature pyramid layer, respectively, as shown in Figure 1. The role of the pyramid level is to solve the scale problem in object detection, and the three pyramid levels of the feature maps are used to predict objects of different scales.

### 2.2. The Optimized FSAF Module

In the FSAF module, IoU loss is adopted for box regression optimization. Compared with IoU loss, CIoU loss can quickly converge and improve performance, so CIoU loss is employed for box regression optimization in this paper, which further optimizes the FSAF module. The basic concept of CIoU loss is introduced in Section 2.2.1.

The weight parameter β is introduced in Section 2.2.2. By reconstructing the loss function of the FSAF module in its online feature selection process, the FSAF module is optimized to enhance the detection performance of the detector for motion-blurred objects.

#### 2.2.1. Adopting CIoU Loss

Generally, when the detector conducts the bounding box regression task, mean square error (MSE) loss and IoU loss can be applied to benefit the IoU metric, but they still suffer from the problems of slow convergence and inaccurate regression. To figure out these problems, CIoU loss is adopted to conduct bounding box regression task, i.e.,
(3)$$L_{CIoU}=1-IoU+\frac{\rho^{2}(b,b^{gt})}{c^{2}}+\alpha \upsilon$$
where b and $b^{gt}$ denote the central points of the predicted box and the target box respectively, IoU is Intersection over Union of b and $b^{gt}$, ρ(⋅) is the Euclidean distance, c is the diagonal length of the smallest enclosing box covering b and $b^{gt}$, α is positive trade-off parameter, and υ measures the consistency of aspect ratio.

Then, the trade-off parameter α is defined as
(4)$$\alpha =\frac{\upsilon }{(1-IoU)+\upsilon }$$
and υ is defined as
(5)$$\upsilon =\frac{4}{\pi^{2}}{\left(\arctan\frac{w^{gt}}{h^{gt}}-\arctan\frac{w}{h}\right)}^{2}$$
where $w^{gt}$ and $h^{gt}$ denote the width and the height of the target box, respectively, and w and h denote the width and the height of the predicted box, respectively.

The reason why we choose CIoU loss as bounding box regression loss of the proposed detector is that CIoU loss considers three important geometric factors, i.e., overlap area, central point distance, and aspect ratio, which will effectively increase the mAP scores of infrared multi-category object detection.

#### 2.2.2. The Weight Parameter β

After a lot of research, we found that the online feature selection loss function of the FSAF module is relatively simple. When the summations of $L_{FL}^{I}(l)$ and $L_{IoU}^{I}(l)$ on each pyramid level of all the anchor-free branches are equal, the FSAF module does not consider which pyramid level will be selected as the best pyramid level to learn the instance. Therefore, the instance cannot be dynamically assigned to the most suitable feature level during training. To overcome this problem, this paper proposes an essential weight parameter β which is adopted to reconstruct the online feature selection loss function, i.e.,
(6)$$L_{l}^{\prime }=\beta L_{FL}^{I}(l)+(1-\beta )L_{IoU}^{I}(l),\;l=1,2,3$$
(7)$$L_{0}^{}=\;\min\{\beta L_{FL}^{I}(l)+(1-\beta )L_{IoU}^{I}(l)\},\;l=1,2,3$$
(8)$$l_{}^{\prime }=\arg\underset{l}{\min}\beta L_{FL}^{I}(l)+(1-\beta )L_{IoU}^{I}(l),\;l=1,2,3$$
(9)$$L^{\prime }=\min\{L_{FL}^{I}(l)+L_{CIoU}^{I}(l)\},\;l=1,2,3$$
(10)$$L_{}^{*}=\min\{\beta L_{FL}^{I}(l)+(1-\beta )L_{CIoU}^{I}(l)\},\;l=1,2,3$$
where β is the weight parameter used to balance classification loss and regression loss in the online feature selection process, $L_{FL}^{I}(l)$ and $L_{IoU}^{I}(l)$ are classification loss and box regression loss on the pyramid level, respectively, I is the instance, l is the pyramid level, $L_{l}^{\prime }$ is the weighted summation of $L_{FL}^{I}(l)$ and $L_{IoU}^{I}(l)$, $L_{0}^{}$ is the weighted minimal summation of $L_{FL}^{I}(l)$ and $L_{IoU}^{I}(l)$, $l^{\prime }$ is the best pyramid level that yields the weighted minimal summation of losses, and its value can be 1, 2, or 3, $L_{CIoU}^{I}(l)$ is the box regression loss of the optimized FSAF module on the pyramid level l, $L_{}^{\prime }$ is the minimal summation of $L_{FL}^{I}(l)$ and $L_{CIoU}^{I}(l)$, and $L^{*}$ is the weighted minimal summation of $L_{FL}^{I}(l)$ and $L_{CIoU}^{I}(l)$.

Figure 2 shows the entire process for online feature selection of the anchor-free branches of the FSAF module with the parameter β. The ground-truth for the classification output is K maps, with each map corresponding to one class. Focal loss is applied for classification supervision with hyperparameters α=0.25 and γ=2.0. The ground-truth for the regression output is four offset maps agnostic to classes. CIoU loss is adopted for box regression optimization in our method.

During training, β is set to different values from 0 to 1, respectively. After many iterations, the value of the parameter β is empirically set to 0.6, and we find that β=0.6 works best in the experiments. When the FSAF module with the weight parameter β learns the best pyramid level of the instance, the weight parameter β=0.6 is set, which is equivalent to increasing the impact proportion of focal loss on the total loss. In other words, when the minimal summations of classification loss and box regression loss on each pyramid level of all anchor-free branches are equal, the pyramid level yielding a smaller classification loss is selected as the best pyramid level for the learning instance; this is because the smaller the classification loss, the better the classification performance of the FASF module.

The proposal of the weight parameter β effectively ensures that each instance of each infrared image can be dynamically assigned to the most suitable feature level during training. For instance, we set β=0.6, when $L_{FL}^{I}(1)=0.5$, $L_{IoU}^{I}(1)=0.5$, $L_{FL}^{I}(2)=0.4$, $L_{IoU}^{I}(2)=0.6$, $L_{FL}^{I}(3)=0.6$, and $L_{IoU}^{I}(3)=0.4$, the values of $L_{1}^{\prime }$, $L_{2}^{\prime }$, and $L_{3}^{\prime }$ are 0.5, 0.48, and 0.52, respectively, by using Equation (6). At this moment, the value of $l^{\prime }$ is 2 by using Equation (8), and the optimized FSAF module will select the second pyramid level as the best level for learning the instance. However, the summations of $L_{FL}^{I}(l)$ and $L_{IoU}^{I}(l)$ on each pyramid level of the FSAF module are equal in the above example, all their values are 1. In this situation, the FSAF module cannot determine the best pyramid level of all anchor-free branches to learn the instance by using Equation (1).

According to the subsequent experimental results, the proposal of the weight parameter β significantly enhances the detection performance of the detector for motion-blurred objects. Besides, it can be seen from Equations (9) and (10) that this paper adopts CIoU loss instead of IoU loss to conduct a series of ablation experiments, which further improves the detection performance of the detector and makes the location of multi-category objects more accurate.

It is worth noting that when the anchor-free branches of the FSAF module with the weight parameter β conduct online feature selection, $L^{*}$ is only the loss employed in selecting the best pyramid level for learning the instance, not the loss used in model training. When training on anchor-free branches, we let $L_{cls}^{af}$ and $L_{reg}^{af}$ be the total classification loss and total regression loss, respectively, then the total loss of the anchor-free branches is $L^{af}=L_{cls}^{af}+L_{reg}^{af}$. For a training batch, features are updated for their correspondingly assigned instances.

### 2.3. Improved Detector Network Architecture

Figure 3 illustrates part of the network architecture of the YOLOv3 detector with the anchor-free branches of the optimized FSAF module. In short, YOLOv3 consists of a backbone network Darknet-53 and some convolutional blocks for specific tasks.

The YOLOv3 detector adopts a similar concept to FPN. Three pyramid levels of the feature maps are used to predict objects of different scales after feature fusion and tensor concatenation. As shown in the anchor-based branch structure of the YOLOv3 detector in Figure 3, it adds several convolutional layers. The last layer of these predicts a three-dimensional (3D) tensor encoding bounding box, objectness, and class predictions. The YOLOv3 detector predicts three boxes at each scale, so the tensor is N×N×[3∗(4+1+K)] for the four bounding box offsets, one objectness prediction, and K class predictions.

In this paper, we add the anchor-free branches of the optimized FSAF module to the 79th, 91st, and 103rd layers of the YOLOv3 network structure, respectively. Limited by the space, Figure 3 only shows that the anchor-free branch is attached to the 91st layer of the YOLOV3 network structure. The anchor-free branches of the improved detector only add two additional convolutional layers for each pyramid layer, shown as the dashed feature map in Figure 3. To be more specific, a 3 × 3 convolutional layer with K filters is attached to the feature map in the anchor-free branch followed by the sigmoid function, in parallel with the one from the anchor-based branch. It predicts the probability of objects at each spatial location for K object classes. Similarly, a 3 × 3 convolutional layer with four filters is attached to the feature map in the anchor-free branch followed by the rectified linear unit (ReLU) [25] function. It is responsible for predicting the box offsets encoded in an anchor-free manner.

Figure 3 illustrates that the improved detector maintains the architecture of the original anchor-based branch. During training and inference, all hyperparameters remain unchanged. Therefore, the anchor-free and anchor-based branches of the improved detector can work jointly in a multi-task manner, sharing the features of each feature pyramid level.

### 2.4. Employing Soft-NMS Algorithm

Generally speaking, non-maximum suppression (NMS) [26] is the last step in most object detection algorithms, in which the redundant detection boxes are removed as long as their overlaps with the highest score box exceed a threshold. For two mutually occluded instances, their detection boxes will inevitably overlap, and NMS may filter out the detection box with the lower score that contains positive instance, resulting in only one instance that can be detected. In this situation, the recall rates of mutually occluded objects are relatively low.

Therefore, the Soft-NMS algorithm instead of NMS is used to select the final detection boxes in the experiment, which enhances the detection performance of the mutually occluded objects. The Soft-NMS algorithm decays the detection scores of all other objects as a continuous function of their overlap with the detection which has the maximum score, i.e.,
(11)$$s_{i}^{*}=s_{i}e^{-\frac{iou{\left(M,b_{i}\right)}^{2}}{\sigma }},\forall b_{i}\notin D$$
where i is the ordinal number of the detection box, $b_{i}$ is the initial detection box, M is the detection box with the maximum score, $iou(M,b_{i})$ is Intersection over Union of M and $b_{i}$, $s_{i}$ is the initial detection score, $s_{i}^{*}$ is the final detection score, σ is the weight parameter, D is the set of final detections, and $e^{-\frac{iou{\left(M,b_{i}\right)}^{2}}{\sigma }}$ is the overlap-based weighting function.

In the subsequent experiments, the value of the parameter σ was set to 0.5.

## 3. Experiments and Discussion

### 3.1. The Infrared Image Dataset

In the experiment, a total of 5322 infrared images were collected through vehicle-mounted infrared thermal imager from dozens of complex scenes, such as campuses, highways, squares, and so on. The image size was 640 × 480 pixels, and the horizontal resolutions and vertical resolutions were both 96 dpi. Referring to the ImageNet Det dataset, MS COCO [27] dataset and PASCAL VOC [28] dataset, we labeled six categories of infrared objects, including car, pedestrian, truck, bike, motorbike, and bus.

The infrared image dataset contains a total of 42,907 labeled instances, and an average image contains about eight labeled instances. Referring to the evaluation metric of the PASCAL VOC dataset, the infrared image dataset uses average precision (AP) with an IoU threshold of 0.5 to evaluate the performance of the dataset. Specifically, for each category in the infrared image dataset, the precision values corresponding to all recall points under the precision-recall curve were averaged to obtain AP, and then the average of AP of all categories was mAP.

### 3.2. Experiment Details

All the experiments were run on a GTXG1080Ti with 11 GB memory, and the Tensorflow framework based on deep learning was used. In the process of training the detection model, we iteratively trained 36k times on our infrared image dataset, where training dataset, validation dataset, and test dataset were 60%, 20%, and 20%, respectively. The initial learning rate was 0.001, which was divided by 10 at 28k and again at 32k iterations. Weight decay was 0.0005 and momentum was 0.9. Furthermore, we conducted the data augmentation operation by using flip transformation, which flips the original image horizontally or vertically.

### 3.3. Ablation Studies

The weight parameter *β* is necessary, it effectively optimizes the FSAF module and overcomes the challenge of the low detection accuracy of the FSAF module for motion-blurred objects. As shown in Table 1, we found that β=0.6 worked best in all the experiments.

However, increasing the value of the weight parameter β will correspondingly reduce the impact proportion of CIoU loss on the total loss. It can be seen from Table 1, 12th entry, that when the value of the parameter β is set to 0.75, the mAP scores are only 54.9%, this is because the impact proportion of CIoU loss on the total loss is so low that the detector cannot overcome the problem of inaccurate regression. Compared with the detector based on the FSAF module, the mAP scores of the detector based on the optimized FSAF module increase by 2.4% (Table 1, 1st vs. 5th entries). When this paper adopts CIoU loss instead of IoU loss to execute the bounding box regression task, the mAP scores of the detector increase by 2.2% (Table 1, 5th vs. 11th entries).

Figure 4 shows that the optimized FSAF module can significantly enhance the detection accuracy of motion-blurred objects. Generally, motion blur will get the important semantic information of the infrared object lost and make it very difficult for the detector to accurately detect the objects, which is reflected in the low object recall and the high false positive rate. The optimized FSAF module increases the proportion of focal loss on the total loss to some extent, and the detection performance of motion-blurred objects is efficiently improved.

CIoU loss is effective, it makes the detection boxes of multi-category objects more accurate in the infrared images. Figure 5 illustrates that the improved detector can greatly improve the precision and recall of infrared multi-category objects when CIoU loss is adopted in the anchor-free and anchor-based branches; it also makes the location of infrared objects more accurate. This is because CIoU loss is more in line with the bounding box regression mechanism than IoU loss and MSE loss, making the bounding box regression more stable.

The Soft-NMS algorithm is significant, it enhances the detection performance of the mutually occluded objects in the thermal infrared images. In complex scenes, pedestrians and vehicles will mutually occlude, and the detection boxes will overlap inevitably. As shown in the green dashed boxes in Figure 6, the original NMS may filter out the detection box with a low score even though it contains a positive instance, resulting in only one object that can be detected. Therefore, NMS will reduce the recall of mutually occluded objects.

As shown in Figure 6, this paper uses the Soft-NMS algorithm instead of NMS to select the final detection boxes, which efficiently increases the detection accuracy of the mutually occluded objects. For those overlapping detection boxes with low detection scores, Soft-NMS will not cause false suppression when the bounding box contains positive instances.

The improved detector proposed in this paper is robust, it enhances the detection performance of the detector for tiny and dense objects in infrared images. We evaluate the contribution of several important elements to our detector, including anchor-based branches, anchor-free branches, CIoU loss, and Soft-NMS. Results are reported in Table 2.

We first trained the YOLOv3 detector and the improved detector, using two efficient strategies including CIOU loss and Soft-NMS, respectively. When jointly optimized with anchor-based branches, the anchor-free branches of the optimized FSAF module help to learn the instances which are hard to be modeled by anchor-based branches, leading to the increased mAP scores (Table 2, 4th, 5th, and 6th entries). Compared with the YOLOv3 detector, the mAP scores of the improved detector increased by 4.1% (Table 2, 3rd vs. 6th entries). To find out what kinds of objects can be detected, our experiment shows some qualitative results of the head-to-head comparison between the YOLOv3 detector and the improved detector in Figure 7.

Clearly, Figure 7 illustrates that the improved detector is better at finding challenging instances, such as tiny and dense objects which are not well covered by anchor boxes.

### 3.4. Comparison to Other Infrared Object Detectors

In this section, the advantage of our improved detector is illustrated by comparing it with other typical object detection methods, where YOLOv2 [29], SSD, deconvolutional single shot detector (DSSD) [30] and YOLOv3 are one-stage detectors, and Faster R-CNN is the two-stage detector. We adopted the pre-trained models provided by these typical object detection methods to train on the infrared image training dataset separately, and then used these trained models to perform image testing on the test dataset. The test results are shown in Table 3.

From Table 3, it can be seen that the proposed method has a mAP score of 72.2%, which is higher than other methods. This effective evaluation metric mAP is due to the strategy that the optimized FSAF module is plugged into the YOLOv3 network structure, which effectively increases the detection accuracy of the detector for tiny, dense, and motion-blurred objects. Besides, the improved detector adopts CIoU loss in the bounding box regression task, which makes the bounding box more accurate and has a high object recall. Finally, the proposed detector uses the Soft-NMS algorithm instead of NMS to select the final detection boxes, which effectively enhances the detection performance of mutually occluded objects.

### 3.5. Discussions

Although the experimental results show that the proposed method has a high detection accuracy, there are still some weaknesses in our study. For instance, compared with the YOLOv3 detector, the proposed detector has lower frames per second (FPS), the FPS of the YOLOv3 detector is 20 (Table 3, 5th entry), while the FPS of the proposed detector is only 18 (Table 3, 6th entry). This is because each anchor-free branch of the optimized FSAF module adds two additional convolutional layers to three feature pyramid layers of the YOLOv3 detector, which will bring 5 ms additional inference time. In addition, after a large number of training iterations, the value of the weight parameter β was empirically set to 0.6 in the experiments.

In future work, some strategies based on reinforcement learning [32] can be tried to determine the optimal value of the parameter β.

It is worth noting that the proposed method has some potential directions for further research. For example, the proposed detector can be trained on some typical optical datasets or remote sensing datasets, such as MS COCO dataset, PASCAL VOC dataset, or DOTA [33] dataset, and the trained detector can detect some typical objects in optical images or remote sensing images. Besides, the proposed method can be used in the direction of moving target tracking for further research.

## 4. Conclusions

This paper proposes a robust thermal infrared vehicle and pedestrian detection method in complex scenes. An important weight parameter β is first proposed to optimize the FSAF module by reconstructing the loss function of the FSAF module in the online feature selection process, which not only increases the detection mAP scores of the infrared objects, but also solves the problem of low detection precision of motion-blurred objects. Then, the optimized anchor-free branches of the FSAF module are plugged into the YOLOv3 single-shot detector, and work jointly with the anchor-based branches of the YOLOv3 detector in both training and inference. This method of combining anchor-based branches and anchor-free branches effectively improves the detection precision of tiny and dense objects. Experimental results show that the proposed method has obvious detection performance, and the mAP scores exceed some typical one-stage and two-stage object detectors.

## Figures and Tables

**Figure 1 sensors-21-01240-f001:**
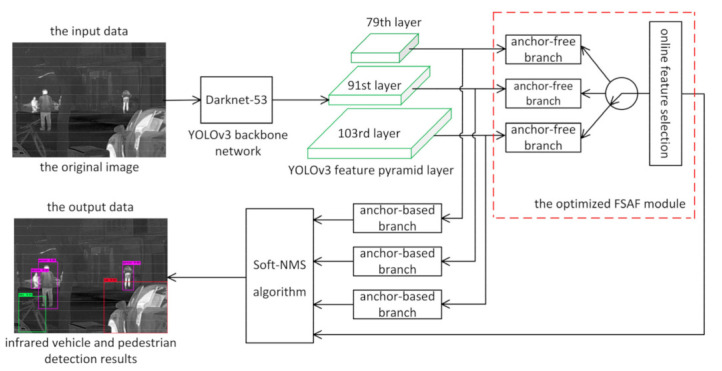
Overview of the proposed method. The optimized feature selective anchor-free (FSAF) module is plugged into the you only look once (YOLO)v3 network structure and works jointly with the anchor-based branches of the YOLOv3 detector.

**Figure 2 sensors-21-01240-f002:**
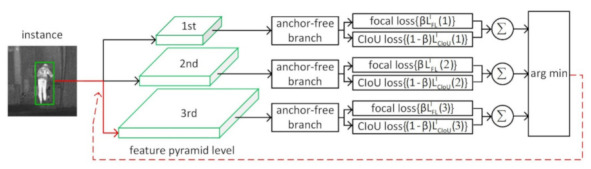
Online feature selection mechanism of the anchor-free branches of the FSAF module with the weight parameter β. Each instance is passing through all levels of anchor-free branches to compute the classification (focal) loss and regression (Complete Intersection over Union (CIoU)) loss. Then, the level with the minimal summation of two losses is selected to set up the supervision signals for learning the instance.

**Figure 3 sensors-21-01240-f003:**
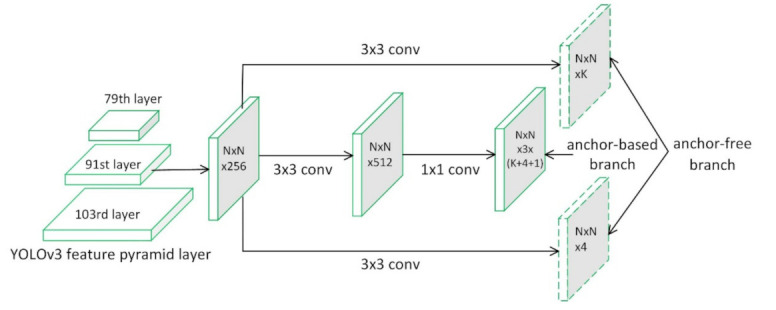
The improved detector network architecture. We plug the optimized FSAF module into the YOLOv3 network structure, which only adds two additional convolutional layers (dashed feature maps) per pyramid level.

**Figure 4 sensors-21-01240-f004:**
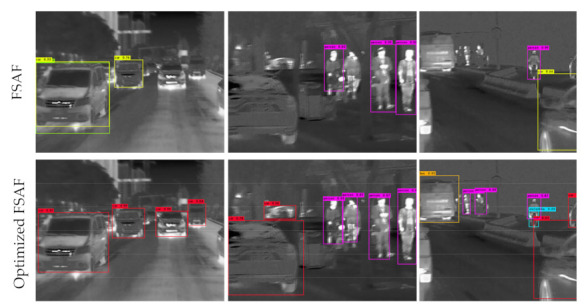
Qualitative test results of the FSAF module and our optimized FSAF module. Both are using DarkNet-53 as the backbone.

**Figure 5 sensors-21-01240-f005:**
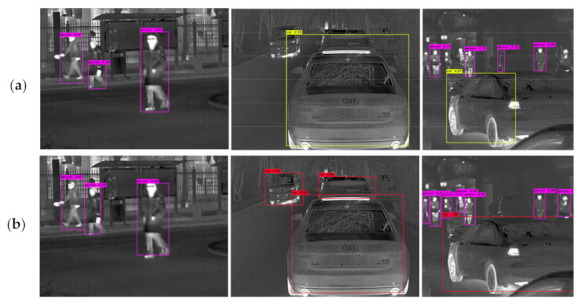
Comparison of the test results of our improved detector using different bounding box regression losses. (**a**) The joint detector with anchor-based branches of YOLOv3 using mean square error (MSE) loss and anchor-free branches of the FSAF module using IoU loss. (**b**) The improved detector using CIoU loss.

**Figure 6 sensors-21-01240-f006:**
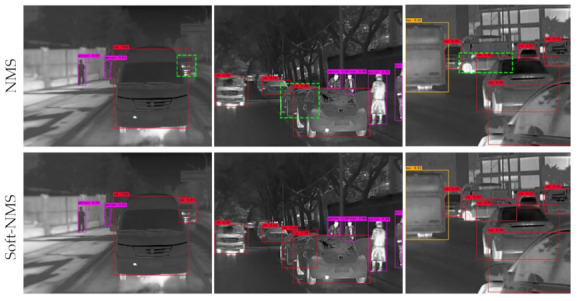
Comparison of the test results of the improved detector using non-maximum suppression (NMS) and the Soft-NMS algorithm, respectively. Both are using DarkNet-53 as the backbone.

**Figure 7 sensors-21-01240-f007:**
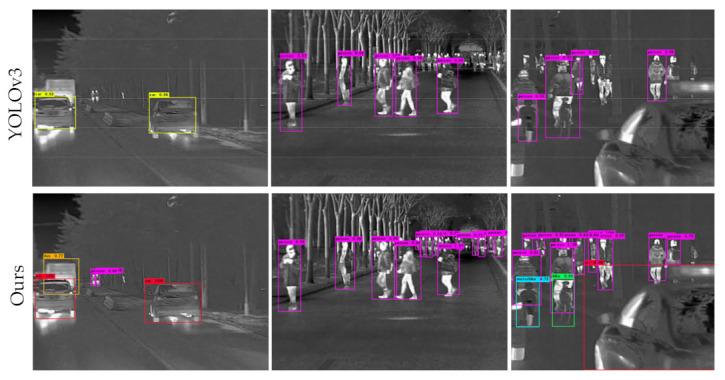
Qualitative results comparison between YOLOv3 detector and the improved detector. Both are using DarkNet-53 as the backbone.

**Table 1 sensors-21-01240-t001:** Detection accuracy with different online feature selection losses on the experiment dataset.

Method	Backbone	Online Feature Selection Loss	β	**mAP (%)**
**FSAF**	DarkNet-53	$L=\underset{}{\min}\{L_{FL}^{I}(l)+L_{IoU}^{I}(l)\}$		66.1
**Ours**	DarkNet-53	$L_{0}^{}=\min\{\beta L_{FL}^{I}(l)+(1-\beta )L_{IoU}^{I}(l)\}$	0.25	52.3
			0.40	63.1
			0.50	66.1
			0.60	68.5
			0.75	52.7
		$L^{\prime }=\min\{L_{FL}^{I}(l)+L_{CIoU}^{I}(l)\}$		68.3
		$L_{}^{*}=\min\{\beta L_{FL}^{I}(l)+(1-\beta )L_{CIoU}^{I}(l)\}$	0.25	54.4
			0.40	64.6
			0.50	68.3
			0.60	70.7
			0.75	54.9

**Table 2 sensors-21-01240-t002:** Ablation experiment of YOLOv3 detector and our improved detector on the infrared image dataset.

Method	Anchor-Based Branches	Anchor-Free Branches	CIoU Loss	Soft-NMS	mAP (%)
YOLOv3	√ √ √		√ √	√	65.8 66.9 68.1
Ours	√ √ √	√ √ √	√ √	√	67.9 70.7 72.2

**Table 3 sensors-21-01240-t003:** Evaluation metric of the improved detector and other typical detectors on the infrared image dataset. AB: Anchor-based branches of the YOLOv3 detector. AF: Anchor-free branches of the improved FSAF module.

Method	Backbone	mAP (%)	FPS
Two-stage:			
Faster R-CNN	ResNet-101 [31]	64.1	2
One-stage:			
YOLOv2	DarkNet-19 [29]	54.1	40
SSD513	ResNet-101	59.6	8
DSSD513	ResNet-101	62.5	6
YOLOv3	Darknet-53 [17]	65.8	20
Ours (AB+AF)	Darknet-53	72.2	18

## Data Availability

Data sharing is not applicable to this article.

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
