# Peer review of "A Robust Thermal Infrared Vehicle and Pedestrian Detection Method in Complex Scenes"

_sensors, 2021, doi:10.3390/s21041240_

Round 1

Reviewer 1 Report

The authors have proposed a modified anchor‐free module for single‐shot object detection. The introduced modification is very simple but establishes a one-sept forward approach. The manuscript cannot be accepted for publication in its current form because of the drawbacks listed below.

The authors should clearly indicate the main contribution of their work in the first Section. The statement "an important weight parameter \beta is proposed for the first time" is not sufficient. Please explain why the weight parameter is essential in the context of the related works. It is necessary to explain what was the motivation for introducing the parameter \beta. The authors' contribution also has to be clarified in the abstract.

The Algorithm in Sect. 2 is not clearly presented. The authors have described the detection algorithm's selected details, but an overview of the method is missing. Please show and explain the algorithm's general structure before presenting details related to its parameters and building blocks. Also, the input and output data of the algorithm should be clearly defined.

Is the source code of the algorithm publicly available? The availability of source code is crucial for replication of the experiments.

My suggestion is to add a section, which would discuss the basic concepts and present the underlying definitions (preliminaries). In such an additional Section, the state-of-the-art FSAF module should be described. Especially, the following terms and concepts have to be explained in details: "heuristic‐guided feature selection", "overlap‐based anchor sampling", "anchor‐free branches", "pyramid level", "instance".

The current organization of the manuscript makes it very difficult to follow. For instance, when reading Section 2.1 the terms "classification loss and regression loss" are not clear for the reader, and the symbols with subscripts IoU and CIoU are confusing. Some information regarding CIoU is provided later in Sect. 2.3. All symbols and definitions used to introduce the modified algorithm should be explained earlier.

Line 133: How the values of the hyper parameters were selected?

What is the meaning of the color of rectangles in Figures 3-6?

Table 1: the column "1 - \beta" is unnecessary.

Section 4: I'm afraid I have to disagree that the mAP of 72.2% can be described as" excellent detection performance". The authors should discuss the weak points of their approach and suggest potential directions for further research.

There are many awkward sentences in the manuscript. For instance, in Abstract, it is stated that "this paper proposes a robust detection method for thermal infrared vehicles and pedestrians in complex scenes", which suggests that there are some "thermal infrared vehicles and pedestrians". In general, the manuscript should be carefully read and corrected by a native speaker.

Author Response

Dear Reviewer,

Thank you very much for taking your time to review this manuscript. We really appreciate all your comments and suggestions! Please find our point-by-point responses in the attachment. Please find our revisions in the resubmitted files. In the resubmitted manuscript, all changes are clearly highlighted in red so that you can easily see them.

Reviewer 2 Report

The contribution of this paper can be summarized as follows.

1. The weight parameter beta is included in the loss function.

2. The soft-NMS is employed for improving the detection performance.

3. An detector network architecture through plugging the optimized FSAF module into the YOLOv3 model.

The research contribution presented in this paper is a little insufficient for a journal paper. Besides, there are several concerns that must be discussed for increasing the readability of the paper. Please consider the following problems or suggestions for revising the manuscript.

1. The weight parameter beta seems a very important factor in the proposed method. However, there is no systematic approach for finding the best value of this parameter. It is hard to convince that the value of 0.6 is good enough to claim that the proposed method is robust.

2. In eq. (6), there are several parameters which should be determined in advance, such as alpha, and the consistency of aspect ratio measurement. They are not described later, and thus, they may be not determined properly. The same issue would be raised in eq. (7).

3. In eq. (7), both sides have the variable of score Si. The presentation is not appropriate, if both sides are divided this score. Subsequently, the exponential part equals to one.

4. From eqs. (2)-(5), all of them are loss function. The notation should be written consistently, using a capital letter L. Besides, the notation of arg{min()} is to find the parameter that minimizes the loss function. It means that the result will be the level of feature pyramid. This situation confuses me what is the output of these equations.

Author Response

(The authors gave the same response as above.)

Round 2

Reviewer 1 Report

The authors have improved the paper by considering all suggestions satisfactory. I believe the paper can be accepted for publication.

Reviewer 2 Report

The authors have revised the manuscript according to the review comments. I think all the point-by-point responses and the revised parts are much better than the previous version.